# Pachychoroid Spectrum Diseases in Patients with Cushing’s Syndrome: A Systematic Review with Meta-Analyses

**DOI:** 10.3390/jcm11154437

**Published:** 2022-07-29

**Authors:** Jeppe K. Holtz, Janni M. E. Larsson, Michael S. Hansen, Elon H. C. van Dijk, Yousif Subhi

**Affiliations:** 1Department of Ophthalmology, Odense University Hospital, DK-5000 Odense, Denmark; jeppe.holtz@rsyd.dk; 2Department of Otolaryngology, Odense University Hospital, DK-5000 Odense, Denmark; 3Department of Ophthalmology, Rigshospitalet, DK-2600 Glostrup, Denmark; janni.margareta.ezban.larsson@regionh.dk (J.M.E.L.); michael.stormly.hansen@regionh.dk (M.S.H.); 4Department of Ophthalmology, Leiden University Medical Center, 2300 RC Leiden, The Netherlands; e.h.c.van_dijk@lumc.nl; 5Department of Clinical Research, University of Southern Denmark, DK-5230 Odense, Denmark

**Keywords:** central serous chorioretinopathy, Cushing’s disease, meta-analysis, pachychoroid spectrum, polypoidal choroidal vasculopathy, systematic review

## Abstract

Cushing’s syndrome is a rare disease with an endogenous cause of excess cortisol secretion. More evidence substantially links cortisol levels to the pachychoroid spectrum diseases. In this systematic review and meta-analysis, we summarize available evidence on pachychoroid spectrum diseases in patients with Cushing’s syndrome. We performed a systematic literature search in 11 databases on 21 May 2022. Studies were considered eligible if they performed retinal examination of a consecutive group of patients with Cushing’s syndrome using optical coherence tomography (OCT) scans. We extracted data on subfoveal choroidal thickness in patients with Cushing’s syndrome compared to matched controls. We also extracted data on the prevalence of pachychoroid pigment epitheliopathy (PPE), central serous chorioretinopathy (CSC), and polypoidal choroidal vasculopathy (PCV). We identified six eligible studies with a total of 159 patients with Cushing’s syndrome. On average, patients with Cushing’s syndrome have 49.5 µm thicker subfoveal choroidal thickness compared to matched healthy individuals. Pachychoroid spectrum diseases were relatively common in these patients: PPE in 20.8%, CSC in 7.7%, and PCV in 2.8%. We conclude that there should be low threshold to recommend ophthalmic examination to patients with Cushing’s syndrome, and that a macular OCT is recommended during this examination.

## 1. Introduction

Cushing’s syndrome is a potentially life-threatening endocrine disease [1,2]. Its etiology is based on endogenous causes for excess cortisol secretion [3]. Most cases are secondary to an adrenocorticotropic hormone (ACTH) producing pituitary tumor. Less commonly, ectopic ACTH producing tumors, tumors releasing the corticotrophin-releasing hormone, or adrenal tumors can cause Cushing’s syndrome [3]. Cushing’s syndrome is rare, and studies estimate a prevalence of 12–62 cases per million and an annual incidence of 1–3 per million [4]. Excess cortisol secretion is considered the underlying cause of the multiple associated co-morbidities such as obesity, dyslipidemia, hypertension, osteoporosis, immune suppression, and neuropsychiatric disturbances [3]. Exogenous Cushing’s syndrome can lead to similar co-morbidities and is typically caused by long-lasting systemic glucocorticoid treatment of various conditions.

Increased cortisol levels, both due to exogenous and endogenous causes, are linked to the development of central serous chorioretinopathy (CSC) [5,6]. Developments in retinal and choroidal imaging, especially during the last decade, have revealed that CSC is linked to a thicker choroid, i.e., pachychoroid, and that this pachychoroid is seen in a spectrum of conditions, which appear to share both clinical nature and etiological findings [7,8,9,10]. It is hypothesized that pachychoroid reflects a choroidal venous overload, which is supported by angiographic evidence of choroidal vascular hyperpermeability, dilated veins, vortex venous anastomoses, and delayed filling of the choroidal veins [7,8,9,10]. These circumstances are hypothesized to explain a disease progression of first a gradual development of a pachychoroid phenotype; followed by a pachychoroid pigment epitheliopathy (PPE), i.e., pigment epithelium detachment (PED) without any subretinal fluid or intraretinal fluid; followed by CSC; and CSC complicated by neovascularizations, e.g., seen as polypoidal choroidal vasculopathy (PCV)/aneurysmal type 1 neovascularization [7,8,9,10].

Considering that patients with Cushing’s syndrome have endogenous causes of excess cortisol secretion, one could explore the choroid and the retina of these patients as a potential disease model for pachychoroid spectrum diseases. To that end, some studies have so far evaluated different aspects of the retina and choroid of patients with Cushing’s syndrome. The aim of this study was to systematically map the literature on findings of pachychoroid spectrum diseases in patients with Cushing’s syndrome.

## 2. Materials and Methods

The systematic review and meta-analysis are reported according to the recommendations stated in the Preferred Reporting Items for Systematic Reviews and Meta-Analyses (PRISMA). The protocol was registered in the PROSPERO database (reg. no. CRD42022335656).

### 2.1. Study Eligibility Criteria

Eligible studies were defined as those which evaluated a consecutive group of patients with Cushing’s syndrome. Studies must perform a retinal examination to evaluate the macula using optical coherence tomography (OCT) scans. Eligible studies must evaluate at least one of the following outcomes: subfoveal choroidal thickness (SFCT) measured using OCT, the prevalence of PPE, CSC, and PCV. While we expected studies to be cross-sectionally observationally designed, we did not enforce any eligibility restrictions on study design and allowed relevant cross-sectional data from any study design deemed otherwise eligible. However, single case studies, publications without original data, conference abstracts, or animal studies were not considered eligible. We only considered studies disseminated in English language.

### 2.2. Information Sources, Search Strategy, and Study Selection

We searched the literature databases PubMed, EMBASE, Web of Science Core Collection, BIOSIS Previews, Current Contents Connect, Data Citation Index, Derwent Innovations Index, KCI-Korean Journal Database, SciELO Citation Index, and the Cochrane Central. One trained author (Y.S.) conducted the search on 21 May 2022. Details of the literature search are available as Appendix A. One author (Y.S.) examined the title and abstracts of all identified records and removed duplicates and those deemed obviously irrelevant. Remaining references were retrieved in full text for the evaluation of eligibility. Two authors (J.H. and J.M.E.L.) then independently examined these full text studies as well as references from these studies for any additional relevant studies. Disagreements between the authors were discussed and if consensus could not be reached, a third author (Y.S.) made the decision on study selection.

### 2.3. Data Extraction and Risk of Bias within Studies

Data on study characteristics, population characteristics, methods for diagnosis, and study results were extracted from each study using data extraction forms. Since we anticipated that most studies would be cross-sectionally designed, risk of bias of individual studies was performed using the relevant items from the Agency for Healthcare Research and Quality (AHRQ) checklist for Cross-Sectional Studies (Questions 1–4 and 6–7), which is the recommended tool for evaluating cross-sectional studies [11]. Two authors (J.H. and J.M.E.L.) independently extracted data and evaluated risk of bias of individual studies. Disagreements between the authors were then discussed and if consensus could not be reached, a third author (Y.S.) made the final decision.

### 2.4. Synthesis of the Results and Risk of Bias across Studies 

Studies were reviewed qualitatively in text and tables and quantitatively using meta-analyses. Meta-analyses were conducted using MetaXL 5.3 (EpiGear International, Sunrise Beach, QLD, Australia) for Microsoft Excel 2013 (Microsoft, Redmont, WA, USA). We used the random-effects model to account for potential heterogeneity in the different design and conduct of the eligible studies in review. For the SFCT, we evaluated the differences in the SFCT between patients with Cushing’s syndrome and healthy controls in the study using the weighted mean difference. For the prevalence of PPE, CSC, and PCV, we performed prevalence meta-analyses. When prevalence numbers reach either 0% or 100%, caution must be shown due to the risk of variance instability and erroneous weighting [12]. Therefore, all prevalence numbers were transformed using the double arcsine method for analysis and back-transformed for interpretation [12]. The Cochran’s Q and I^2^ were calculated to quantify heterogeneity in our estimates [13]. Funnel plots were used to evaluate skewed results and publication bias [14]. Sensitivity analysis was conducted by removing each study in turn and re-calculating the summary measures to evaluate the magnitude of the change in the results.

## 3. Results

### 3.1. Literature Search and Study Selection

The literature search identified 82 records in total. After discarding duplicates (N = 23) and obviously irrelevant records (N = 49), 10 records remained for full-text evaluation. Of these, three records were excluded as they were case reports, and one record was excluded as it investigated the prevalence of Cushing’s syndrome in patients with CSC and was therefore deemed as the wrong population of interest. Finally, six records were included for our review (Figure 1).

### 3.2. Study Characteristics

Eligible studies included a total of 159 patients with Cushing’s syndrome [15,16,17,18,19,20]. Five studies also included healthy controls (a total of 159 individuals) for comparison [15,17,18,19,20]. All studies were cross-sectional in design. Studies were based on populations from Brazil, China, France, Italy, Netherlands, and Turkey. Further details on study populations and study definitions and diagnosis of Cushing’s syndrome are listed in detail in Table 1. 

Ophthalmic examination of the patients included slit-lamp biomicroscopy [15,17,18,19,20], indirect ophthalmoscopy [15,16,18,20], and fundus photography [16,17]. All studies had retinal OCT, which was also an eligibility requirement for inclusion in this review. Pupillary dilation was reported by one study [16]. Retinal angiography was methodologically described by two studies [17,19]; however, remaining studies described or discussed cases with retinal pathology diagnosed using retinal angiography. Five studies used spectral domain OCT [15,16,17,18,19], and one study used swept source OCT [20]. Studies of spectral domain OCT all employed enhanced depth imaging (EDI) protocol [15,16,17,18,19], in which the device is placed closer to the eye than regularly which focuses the illumination at the level of choroid or inner sclera and allows better level of detail at the level of the choroid. The EDI OCT allows outlining of the choroid and measurement of the SFCT. The SFCT was measured in all studies [15,16,17,18,19,20], the prevalence of PPE was reported in three studies [15,17,19], the prevalence of CSC was reported in all studies [15,16,17,18,19,20], and the prevalence of PCV was reported in three studies [15,17,19]. Further details of the ophthalmic examination protocol are described in Table 2. 

Patients with Cushing’s syndrome had a mean age ranging from 38 to 53 years. Females constituted the majority of patients with Cushing’s syndrome. In studies with healthy control participants, the age and gender distribution were similar to that of the patient group. Disease duration ranged from recently diagnosed cases to cases with a duration of 30 years. In terms of etiology of cases with Cushing’s syndrome, pituitary adenomas constituted the majority. Details of participant characteristics are outlined in Table 3.

### 3.3. Results of Individual Studies

Abalem et al. evaluated retinal OCTs to compare choroidal thickness in patients with Cushing’s syndrome and to compare to healthy controls [15]. The authors found that there was a significantly thicker choroid in eyes of patients with active Cushing’s syndrome, and that 18% of patients also presented with macular changes assumed to be secondary to the thicker choroid [15]. Brinks et al. systematically examined eyes of patients with Cushing’s syndrome and found that CSC and other macular abnormalities were common, with three of eleven patients having CSC but also two patients having hypertensive retinopathy, and suggest a low threshold of ophthalmic evaluation of patients with Cushing’s disease [16]. Eymard et al. examined patients with Cushing’s syndrome and found that 21% had any features of pachychoroid spectrum disease [17]. The authors also investigated associates of the SFCT and found that age, spherical equivalent, cumulative disease duration, and pituitary etiology were all factors of significant influence, whereas neither mean 24-h urinary free cortisol at time of evaluation nor 24-h urinary free cortisol before commencement of treatment for Cushing’s syndrome were significant factors on the SFCT [17]. Karaca et al. reported that patients with Cushing’s syndrome had significantly larger choroidal thickness at multiple examination points when compared to healthy controls, and that the choroidal thicknesses were correlated with adrenocorticotropic hormone levels [18]. Lassandro et al. found that patients with Cushing’s syndrome had a significantly larger SFCT and higher values of choriocapillaris vessel density [19]. Further, more than one in three patients presented with any pachychoroid spectrum feature [19]. Wang et al. found a significantly greater SFCT in patients with Cushing’s syndrome when compared to healthy controls, and that 53% were defined as having pachychoroid [20]. SFCT was significantly correlated with 24-h urinary free cortisol, but neither plasma-free cortisol nor adrenocorticotropic hormone levels [20].

### 3.4. Risk of Bias of Individual Studies

Risk of bias evaluation of the individual studies showed that studies clearly reported source of data, eligibility criteria of participants, and time period of investigation. Recruitment process was clearly outlined in four studies, and unclear in two studies. All studies apart from one had two investigators to evaluate images of the participants. Where exclusions were made, studies clearly explained rationale for such exclusions. Table 4 outlines the risk of bias assessment of individual studies.

### 3.5. Synthesis of Results: Difference in Subfoveal Choroidal Thickness between Eyes from Patients with Cushing’s Syndrome and Healthy Controls 

For this analysis, we extracted data from five studies with data on a total of 439 eyes. The random-effects summary estimate of the weighted mean difference was 49.5 µm (95% confidence interval: 10.1–89.0 µm, *p* = 0.014) (Figure 2). Heterogeneity statistics found substantial heterogeneity (Cochran’s Q = 21.6; *p* < 0.01; I^2^ = 82%). The Funnel plot did not suggest risk of bias across studies (Appendix A). Sensitivity analysis showed overall robustness of the summary estimate with persisting direction of results; however, removing Wong et al. [20] in the sensitivity analysis led to loss of statistical significance (Appendix A). Additionally, the sensitivity analysis also illustrates that the heterogeneity is reduced markedly when either Abalem et al. [15] or Eymard et al. [17] are removed from the analysis (Appendix A).

### 3.6. Synthesis of Results: Prevalence of Pachychoroid Pigment Epitheliopathy, Central Serous Chorioretinopathy, and Polypoidal Choroidal Vasculopathy in Eyes from Patients with Cushing’s Syndrome and Healthy Controls

This part of the analysis is based on three separate prevalence meta-analyses on the prevalence of PPE, CSC, and PCV in patients with Cushing’s syndrome. 

For the prevalence of PPE, we extracted data from three studies with data on 142 eyes. The random-effects summary prevalence estimate was 20.8% (95% confidence interval: 8.7–36.1%) (Figure 3). Heterogeneity statistics found substantial heterogeneity (Cochran’s Q = 7.3; *p* = 0.02; I^2^ = 73%). 

For the prevalence of CSC, we extracted data from five studies with data on 198 eyes. The random-effects summary prevalence estimate was 7.7% (95% confidence interval: 1.0–18.7%) (Figure 4). Heterogeneity statistics found substantial heterogeneity (Cochran’s Q = 19.6; *p* < 0.01; I^2^ = 80%).

For the prevalence of PCV, we extracted data from four studies with data on 164 eyes. The random-effects summary prevalence estimate was 2.8% (95% confidence interval: 0.7–6.0%) (Figure 5). Heterogeneity statistics found minimal heterogeneity (Cochran’s Q = 1.3; *p* = 0.74; I^2^ = 0%).

Risk of bias across studies evaluated with Funnel plots was not suggestive of risk of bias across studies, although small number of studies in some analyses (<5 studies) should be acknowledged when interpreting the results (Appendix A). Sensitivity analyses of the prevalence meta-analyses showed overall robustness of the summary estimates (Appendix A).

## 4. Discussion

In this systematic review and meta-analysis, we systematically evaluated the existing literature on any pachychoroid spectrum disease findings in patients with Cushing’s syndrome. Our weighted mean summary estimates find that on average, patients with Cushing’s syndrome have 49.5 µm thicker subfoveal choroidal thickness than matched healthy individuals. Beyond the thicker choroid, it also appears that the prevalence of any pachychoroid spectrum disease is high in this patient group. Approximately one in five patients with Cushing’s syndrome may have PPE, and the prevalence of CSC is 7.7% and PCV is 2.8%. For comparison, prevalence studies find these conditions at a much lower rate in the general population: the prevalence of CSC is estimated to approximately 0.1% in elderly Chinese in the Beijing Eye Study [21], and the prevalence of PCV is estimated to approximately 0.04% [22]. These findings of our review strongly suggest that pachychoroid spectrum diseases are significantly more common in patients with Cushing’s syndrome.

The choroid is a complex vasculature and regulation of the choroidal structure and perfusion has many contributors: neuronal signaling, regulation according to metabolic demand, potential regulation by the intraocular pressure, age-related changes of the macula and the choroid, and the chorioretinal immunity [23,24,25,26]. Interestingly, emerging evidence suggests that cortisol and glucocorticoid receptors play a significant part in the regulation of one or more of these factors. Brinks et al. evaluated post-mortem donor choroids and found that the glucocorticoid receptor is highly expressed in the human choroid [27]. Using RNA-sequencing on isolated donor choroidal endothelial cells, the authors reported that cortisol stimulation led to changes in RNA-expression, interfering with pathways linked to hypertension, fibrosis, vascular permeability, and choroidal neovascularization [27]. 

Limitations of our study need to be acknowledged when interpreting its results. First, included studies did not have a homogenous definition of the different disease entities, which may contribute inaccuracies to the summary estimates. Second, our review is based on 159 patients with Cushing’s syndrome, which at initial glance seems like a small number of patients. However, considering that this is a rare disease [1,2], and that individual studies were only able to include 11–49 patients, our summary of the retinal status in 159 patients with Cushing’s syndrome can be considered a large number. In any case, it reflects the entirety of what is known today. Finally, although the inclusion was consecutive, referral patterns to tertiary research hospitals which are more likely to conduct these studies, may influence the severity of the diseases seen and may therefore also provide a bias towards higher prevalence of patients with any pachychoroid spectrum disease.

In conclusion, we find that patients with Cushing’s syndrome have a high prevalence of any pachychoroid spectrum disease. Thus, in patients with Cushing’s syndrome there should be generally low threshold to recommend or to refer to an ophthalmic examination. When performing ophthalmic examination in patients with Cushing’s syndrome, it is recommended to perform a retinal and a macular examination, which as a minimum should include a macular OCT. This is important as a diagnosis of CSC could need treatment, and PCV or any macular neovascularization should undergo timely treatment to avoid untreatable fibrosis with irreversible vision loss [28,29].

## Figures and Tables

**Figure 1 jcm-11-04437-f001:**
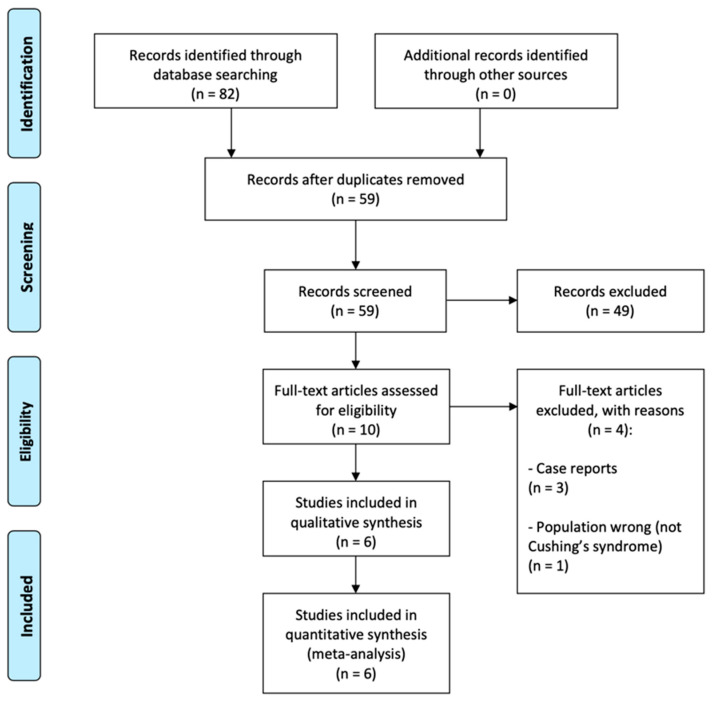
PRISMA flow diagram of study selection.

**Figure 2 jcm-11-04437-f002:**
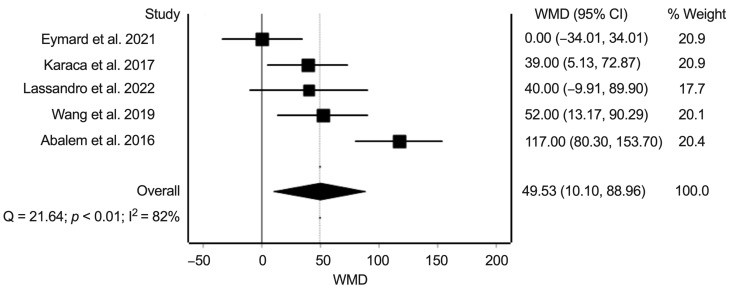
Meta-analysis of the weighted mean difference (WMD) in µm in the subfoveal choroidal thickness between eyes of patients with Cushing’s syndrome and eyes of matched healthy controls. Heterogeneity statistics (Cochran’s Q = 21.6; *p* < 0.01; I^2^ = 82%) show substantial heterogeneity across individual studies [15,17,18,19,20].

**Figure 3 jcm-11-04437-f003:**
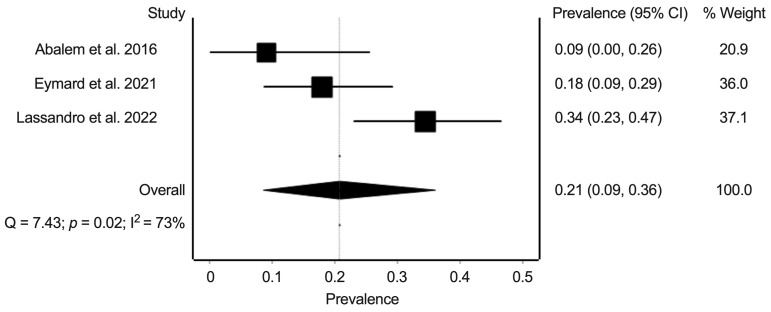
Meta-analysis of the prevalence of pachychoroid pigment epithelium in eyes of patients with Cushing’s syndrome. Prevalence numbers are indicated in decimals [15,17,19].

**Figure 4 jcm-11-04437-f004:**
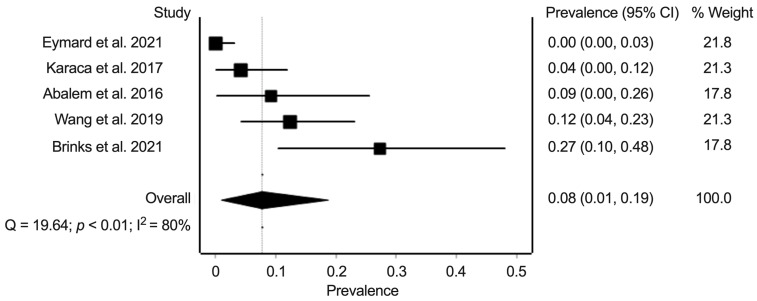
Meta-analysis of the prevalence of central serous chorioretinopathy in eyes of patients with Cushing’s syndrome. Prevalence numbers are indicated in decimals [15,16,17,18,20].

**Figure 5 jcm-11-04437-f005:**
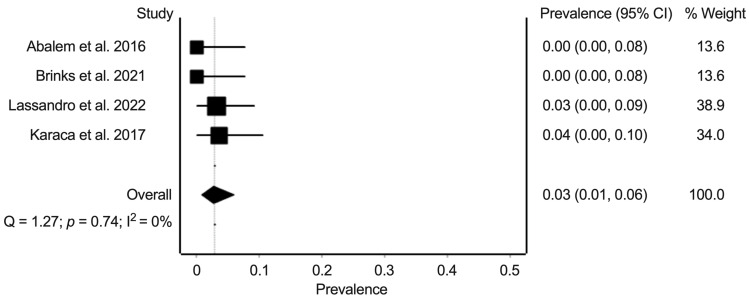
Meta-analysis of the prevalence of polypoidal choroidal vasculopathy in eyes of patients with Cushing’s syndrome. Prevalence numbers are indicated in decimals [15,16,18,19].

**Table 1 jcm-11-04437-t001:** Study characteristics of eligible studies.

Reference	Country	Study Design	Study Population	Definition of Cushing’s Syndrome
Abalem et al., 2016 [15]	Brazil	Cross-sectional	Patients with active Cushing’s syndrome. Healthy controls were recruited for comparison. Participants were excluded if refraction of more than ±6.0 D spherical equivalent, presence of >2.0 D astigmatism, axial length > 26.5 mm, media opacities, history of uveitis or retinal diseases other than that seen in the pachychoroid spectrum, glaucoma or any other optic neuropathy, recent intraocular surgery, or any previous intravitreal injection or laser treatment.	≥2 abnormal screening tests (insufficient suppression of cortisol after low dose dexamethasone suppression test, increased salivary cortisol, or increased 24-h urinary free cortisol).
Brinks et al., 2021 [16]	Netherlands	Cross-sectional	Patients with active Cushing’s syndrome either as a first episode or a new episode in case of recurrence of disease. Eligible patients did not have any visual complaints at the time of establishing the diagnosis.	≥2 abnormal screening tests (insufficient suppression of cortisol after low dose dexamethasone suppression test, increased salivary cortisol, or increased 24-h urinary free cortisol). Etiology of the disease was assessed by MRI or CT.
Eymard et al., 2021 [17]	France	Cross-sectional	Patients with Cushing’s syndrome. Healthy controls were recruited for comparison. Participants were excluded if any other retinal disease (including diabetic retinopathy, age-related macular degeneration, myopia > 6 D, macular telangiectasia, retinal artery/vein occlusion) or a poor image quality.	Hormone testing (not specified in detail). Etiology of the disease was assessed by imaging.
Karaca et al., 2017 [18]	Turkey	Cross-sectional	Patients with newly diagnosed Cushing’s syndrome. Healthy controls were recruited for comparison. Participants were excluded if any previous photodynamic therapy, intravitreal injections, posterior segment surgery, myopia > 6 D or axial length > 26.5 mm, amblyopia, glaucoma, macular degeneration, proliferative retinopathy, uncontrolled diabetes, hypertension, or poor OCT quality.	Low-dose dexamethasone suppression test, midnight serum cortisol test, adrenocorticotropic hormone test, and dehydroepiandrosterone sulfate test. Etiology of the disease was assessed by imaging.
Lassandro et al., 2022 [19]	Italy	Cross-sectional	Patients with active and not active Cushing’s syndrome. Healthy controls were recruited for comparison. Participants were excluded if refraction of more than ±6.0 D spherical equivalent, history of glaucoma, uveitis, or retinal diseases, history of eye surgery, other systemic diseases except for controlled secondary hypertension in patients with Cushing’s syndrome, or media opacities.	≥2 abnormal screening tests (insufficient suppression of cortisol after low-dose dexamethasone suppression test, increased salivary cortisol, or increased 24-h urinary free cortisol).
Wang et al., 2019 [20]	China	Cross-sectional	Patients with active Cushing’s syndrome either as a first episode or a new episode in case of recurrence of disease. Healthy controls were recruited for comparison. Participants were excluded if aged < 18 years, refraction of more than ±6.0 D spherical equivalent, axial length > 26.5 mm, history of glaucoma, uveitis, or retinal diseases except for serous chorioretinopathy, history of laser, intravitreal injection or ocular surgery, systemic diseases except for secondary hypertension or secondary diabetes in patients with Cushing’s syndrome, history of exogenous glucocorticoid exposure, or media opacities.	Presence of cushingoid appearance, failure to achieve midnight nadir in cortisol diurnal rhythm, lack of negative feedback in low-dose dexamethasone suppression test, increased excretion of urine-free cortisol, and imaging of the pituitary and the adrenal gland. All patients with Cushing’s syndrome had plasma-free cortisol, 24-h urinary free cortisol, and plasma adrenocorticotropic hormone test.

Abbreviations: CT = computed tomography; D = diopters; MRI = magnetic resonance imaging; OCT = optical coherence tomography.

**Table 2 jcm-11-04437-t002:** Methods of ophthalmic examination.

Reference	General Ophthalmic Examination	OCT Protocol	Outcomes of Interest for This Review
Abalem et al., 2016 [15]	Slit-lamp biomicroscopy, indirect fundoscopy, axial length measurement, and retinal OCT. Pupillary dilation was not reported.	SD-OCT (Spectralis, Heidelberg Engineering) with an EDI protocol (horizontal and vertical scans, seven sections, high resolution mode, 25 frames). All OCTs were obtained at the same time (without further specification).	SFCT was measured. Patients with any PPE, CSC, and PCV were reported.
Brinks et al., 2021 [16]	Indirect ophthalmoscopy, fundus photography, fundus autofluorescence, fluorescein angiography, and retinal OCT. Pupils were dilated.	SD-OCT (Spectralis HRA + OCT, Heidelberg Engineering) with an EDI protocol.	SFCT was measured. Presence of any CSC was reported.
Eymard et al., 2021 [17]	Slit-lamp biomicroscopy, fundus photography, fundus photography, fundus autofluorescence, and retinal OCT. Pupillary dilation was not reported.	SD-OCT (Spectralis, Heidelberg Engineering) with an EDI protocol and OCT-angiography (OCT-A, RTVue XR Avanti, Optovue Inc.). EDI scans were all performed between 2 pm and 5 pm.	SFCT was measured. Presence of any PPE, CSC, and PCV were reported.
Karaca et al., 2017 [18]	Slit-lamp biomicroscopy, indirect fundoscopy, axial length measurement, and retinal OCT. Pupillary dilation was not reported.	SD-OCT (Spectralis, Heidelberg Engineering) with an EDI protocol (horizontal scans, seven sections, 100 averaged images).	SFCT was measured. Presence of any CSC was reported.
Lassandro et al., 2022 [19]	Slit-lamp biomicroscopy, retinal OCT, and in select cases also fluorescein and indocyanine green angiography. Pupillary dilation was not reported.	SD-OCT (RS-3000 Advance 2, Nidek Co., Ltd.) with an EDI protocol. OCT scans were all performed between 1 pm and 5 pm.	SFCT was measured. Presence of any PPE, CSC, and PCV were reported.
Wang et al., 2019 [20]	Slit-lamp biomicroscopy, indirect ophthalmoscopy, axial length measurement, and retinal OCT. Pupillary dilation was not reported.	SS-OCT (DRI OCT Triton plus, Topcon Corp.). All scans were performed in the afternoon.	SFCT was measured. Presence of any CSC was reported.

Abbreviations: CSC = central serous chorioretinopathy; EDI = enhanced depth imaging; OCT = optical coherence tomography; PCV = polypoidal choroidal vasculopathy; PPE = pachychoroid pigment epitheliopathy; SFCT = subfoveal choroidal thickness.

**Table 3 jcm-11-04437-t003:** Participant characteristics.

Reference	*Patients with Cushing’s Syndrome*					*Healthy Controls*		
	N	Age	Females	Etiology	Duration of Disease	N	Age	Females
Abalem et al., 2016 [15]	11	38 ± 16	100%	8 pituitary adenoma, 1 adrenocortical adenoma, 1 adrenocortical carcinoma, 1 primary macronodular adrenal hyperplasia.	5–360 months	12	51 ± 17	100%
Brinks et al., 2021 [16]	11	53 ± 16	64%	7 pituitary adenoma, 3 adrenal adenoma, 1 undetermined.	1–25 weeks	—		
Eymard et al., 2021 [17]	28	47 ± 15	82%	19 pituitary adenoma, 4 primary macronodular adrenal hyperplasia, 2 adrenocortical carcinoma, 2 ectopic ACTH secretion, 1 undetermined.	—	28	46 ± 12	82%
Karaca et al., 2017 [18]	28	43 ± 13	75%	16 ACTH secreting pituitary adenoma, 10 unilateral cortisol-producing adenoma, 2 bilateral macronodular adrenal hyperplasia	—	38	44 ± 12	76%
Lassandro et al., 2022 [19]	32	Median 48	84%	—	Mean 95 months	32	Median 48	86%
Wang et al., 2019 [20]	49	41 ± 12	88%	44 pituitary adenoma, 5 adrenal gland adenoma	—	49	41 ± 12	88%

Abbreviations: ACTH = adrenocorticotropic hormone; N = number.

**Table 4 jcm-11-04437-t004:** Risk of bias within individual studies included in the review.

Reference	Defines Source	Eligibility Criteria	Time Period	Consecutive Recruitment	Quality Assurance	Explains Exclusions
Abalem et al., 2016 [15]	Yes	Yes	Yes	Unclear	Yes	Not relevant
Brinks et al., 2021 [16]	Yes	Yes	Yes	Yes	Yes	Not relevant
Eymard et al., 2021 [17]	Yes	Yes	Yes	Yes	Yes	Not relevant
Karaca et al., 2017 [18]	Yes	Yes	Yes	Unclear	No	Yes
Lassandro et al., 2022 [19]	Yes	Yes	Yes	Yes	Yes	Yes
Wang et al., 2019 [20]	Yes	Yes	Yes	Yes	Yes	Yes

Studies are assessed on relevant items from the Agency for Healthcare Research and Quality checklist: Defines source: Defines the source of information. Eligibility criteria: Lists inclusion and exclusion criteria or refers to previous publications. Time period: Indicates time period used for identifying participants. Consecutive recruitment: Indicates whether or not subjects were consecutively recruited for the study. Quality assurance: Describes any assessments undertaken for quality assurance purposes. Explains exclusions: Explains any patient exclusions from analysis.

## Data Availability

Not applicable.

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
