# Peer review of "Pachychoroid Spectrum Diseases in Patients with Cushing’s Syndrome: A Systematic Review with Meta-Analyses"

_jcm, 2022, doi:10.3390/jcm11154437_

Round 1

Reviewer 1 Report

This review article focuses on pachychoroid spectrum diseases in patients with Cushing's syndrome. The paper covers the topic which has become an important area of research recently. The manuscript is correct in terms of the structure and the merits. Material and methods are clearly described , statistical analysis is performed appropriately and rigorously, and the conclusions are supported by the data. The results are valuable and will be interesting for the specialists.

Author Response

Reviewer #1:

This review article focuses on pachychoroid spectrum diseases in patients with Cushing's syndrome. The paper covers the topic which has become an important area of research recently. The manuscript is correct in terms of the structure and the merits. Material and methods are clearly described , statistical analysis is performed appropriately and rigorously, and the conclusions are supported by the data. The results are valuable and will be interesting for the specialists.

Authors’ response:

Thank you for your time and comments.

Reviewer 2 Report

This systematic review represents an interesting topic in Ophthalmology and Endocrinology/Neurology. It is well-structured and nicely elaborated in terms of content. The statistics are clear (except for the patient numbers in the calculation of the prevalence). I only have a few comments for adjustments:

Introduction:

When you use exogenous / endogenous Cushing's syndrome as technical terms in your manuscript you should at least shortly mention and describe the exogenous type in the first paragraph as well.

Table 1:

For the ease of the reaser it would be better to sum up the inclusion/exclusion criteria as bullet points and not as continuous text.

e.g.

- patients with active Cushing vs. healthy controls

- exclusion criteria: ...

M&M:

Please shortly explain what EDI-OCT is and why it is better suitable for analysing the choroid than SD-OCT (especially for the non-Ophthalmologists).

Results:

"Brinks et al. systematically examined eyes of patients with 171 Cushing’s syndrome and found that CSC and other macular abnormalities were common".

Please state the exact prevalence in Brinks study instead of saying macular abnormalities were "common".

Prevalence/Figure 4-5:

How come you calculate the prevalence for CSC/PCV in 198, rsp. 164 patients with Cushing but only 159 patients with Cushing were included in the meta analysis?

Figure 5:

Adjust x-axis/abscissa

Discussion:

"...it is recommended to perform a retinal and a macular examination, which as a minimum should include a macular OCT."

Since you suggest every patient with Cushing syndrome should get an ophthalmological exam and an OCT it should also be outlined if these pachychoroid spectrum diseases cause symptoms and what the consequences would be. Usually, as far as I know, the therapy in PPE and symptom-free CSC is to stabilize the cortisol blood levels/watch and wait (in patients without symptoms/visual impairment).  Otherwise critically discuss this suggestion for only patients with ophthalmological symptoms.

Author Response

Reviewer #2 general comments:

This systematic review represents an interesting topic in Ophthalmology and Endocrinology/Neurology. It is well-structured and nicely elaborated in terms of content. The statistics are clear (except for the patient numbers in the calculation of the prevalence). I only have a few comments for adjustments:

Authors’ response:

Thank you for your time and comments.

Reviewer #2 comment #1:

Introduction:

When you use exogenous / endogenous Cushing's syndrome as technical terms in your manuscript you should at least shortly mention and describe the exogenous type in the first paragraph as well.

Authors’ response:

We have now shortly described exogenous Cushing's syndrome in the first paragraph as suggested.

Reviewer #2 comment #2:

Table 1:

For the ease of the reaser it would be better to sum up the inclusion/exclusion criteria as bullet points and not as continuous text.

e.g.

- patients with active Cushing vs. healthy controls

- exclusion criteria: ...

Authors’ response:

Thank you for this valuable suggestion. We have now revised the Table accordingly.

Reviewer #2 comment #3:

M&M:

Please shortly explain what EDI-OCT is and why it is better suitable for analysing the choroid than SD-OCT (especially for the non-Ophthalmologists).

Authors’ response:

Thank you for this suggestion. This is an important point, which we now outline in greater detail as recommended. 

Reviewer #2 comment #4:

Results:

"Brinks et al. systematically examined eyes of patients with 171 Cushing’s syndrome and found that CSC and other macular abnormalities were common".

Please state the exact prevalence in Brinks study instead of saying macular abnormalities were "common".

Authors’ response:

We have now added exact prevalence numbers from the Brinks study.

Reviewer #2 comment #5:

Prevalence/Figure 4-5:

How come you calculate the prevalence for CSC/PCV in 198, rsp. 164 patients with Cushing but only 159 patients with Cushing were included in the meta analysis?

Authors’ response:

We have 159 patients in our review of all studies in total. Some studies included one eye per participant, some studies included both eyes. For each outcome of interest, we could only include studies that had looked at that specific outcome. So, for the prevalence of CSC, five studies with data on 198 eyes were eligible for analysis. For the prevalence of PCV, four studies with data on 164 eyes were eligible for analysis. For the sake of the readership, we have already outlined this as "patients" when we refer to number of patients, and "eyes", when we refer to analysis of eyes.

Reviewer #2 comment #6:

Figure 5:

Adjust x-axis/abscissa.

Authors’ response:

The x-axis of figures 3, 4, and 5 are similar for better comparison. We believe that it allows the reader to clearly see the prevalence gradient across the severity of pachychoroid disease spectrum. We hope that the reviewer can agree on this point given this explanation.

Reviewer #2 comment #7:

Discussion:

"...it is recommended to perform a retinal and a macular examination, which as a minimum should include a macular OCT."

Since you suggest every patient with Cushing syndrome should get an ophthalmological exam and an OCT it should also be outlined if these pachychoroid spectrum diseases cause symptoms and what the consequences would be. Usually, as far as I know, the therapy in PPE and symptom-free CSC is to stabilize the cortisol blood levels/watch and wait (in patients without symptoms/visual impairment).  Otherwise critically discuss this suggestion for only patients with ophthalmological symptoms.

Authors’ response:

Thank you for this point. To be clear, the citation does not reflect the complete sentence and point, which states "Thus, in patients with Cushing’s syndrome there should be generally low threshold to recommend or to refer to an ophthalmic examination. When performing ophthalmic examination in patients with Cushing’s syndrome, it is recommended to perform a retinal and a macular examination, which as a minimum should include a macular OCT".

Thus, we do not recommend that all patients with Cushing's syndrome should get an ophthalmological exam or an OCT. We recommend that there should be generally low threshold to recommend or to refer to an ophthalmic examination.

Regardless, we agree with the general point of the reviewer, that a sentence regarding consequences is appropriate. Thus, we now state that a diagnosis of CSC can lead to treatment consideration, and that PCV or any type of macular neovascularization should undergo timely treatment to avoid untreatable fibrosis with irreversible vision loss.